# Influence of Climate Change on the Thermal Condition of Yakutia's Permafrost Landscapes (Chabyda Station)

**Stepan P Varlamov \*, Yuri B Skachkov and Pavel N Skryabin**

Melnikov Permafrost Institute SB RAS, 677010 Yakutsk, Russia

**\*** Correspondence: svarlamov@mpi.ysn.ru

**Abstract:** This paper presents the results of 39 years of observations conducted at the Chabyda station to monitor the thermal state of permafrost landscapes under current climatic warming. The analysis of long-term records from weather stations in the region has revealed one of the highest increasing trends in mean annual air temperature in northern Russia. The partitioning of the energy balance in different landscape units within the study area has been analyzed. Quantitative relationships in the long-term variability of ground thermal parameters, such as the ground temperature at the bottom of the active layer and seasonal thaw depth, have been established. The ground temperature dynamics within the depth of zero annual amplitude indicates that both warm and cold permafrost are thermally stable. The short-term variability of the snow accumulation regime is the main factor controlling the thermal state of the ground in permafrost landscapes. The depth of seasonal thaw is characterized by low interannual variability and exhibits little response to climate warming, with no statistically significant increasing or decreasing trend. The results of the ground thermal monitoring can be extended to similar landscapes in the region, providing a reliable basis for predicting heat transfer in natural, undisturbed landscapes.

**Keywords:** seasonal thaw depth; climate change; energy balance; monitoring; permafrost response; ground temperature

## 1. Introduction

In the context of research into global climatic change, interest in the problems of the response of permafrost to these changes has increased. Research on the impact of climate warming on permafrost involves a wide variety of issues, including the thermal evolution of upper permafrost in natural and disturbed landscapes under anthropogenic effects. This research has become a priority in geocryological science, with theoretical and practical significance worldwide.

In the face of the major climate changes in recent decades in Central Yakutia [1]—a very densely populated and promising region in terms of agricultural and industrial development—obtaining information on the response of permafrost to modern warming is critical.

Pioneering systematic observations of temperatures in 10–15 m depth bore holes were conducted in 1935. The study revealed peculiarities of temperature variations and ground thaw depth under the effects of vegetation and snow cover, the "cultural layer", and geological and geomorphological parameters [2,3]. In the 1940s, year-round experimental observations were conducted to study ground temperature regime variations under thermal insulation, snow and vegetation cover removal, and in natural landscapes of experimental sites of the Yakut permafrost research station [4–6]. In the 1950s–1960s, more detailed integrated thermal physics observations of the shallow soil formation thermal regime were performed, applying updated methods at experimental sites of the North-Eastern department of the Permafrost Institute [7].

Local and regional patterns of the formation of the thermal regime of soils are revealed in the most detailed way with the wide application of seasonal stationary research methods [8,9]. In the 1960s–1980s, regular observations of the thermal balance according to a significantly updated program were conducted at Yakutsk, Syrdakh, Zelenyi Lug and Chabyda stations [10–13]. The research covered both natural and disturbed landscapes. Daily, seasonal and annual variations in the surface energy balance were studied in detail, and a series of new mathematical models of ground thawing and warming were elaborated. Additionally, hydrothermal parameters of frozen ground under the impact of agricultural and reclamation projects were studied at Khatassy, Khorobut, and Amga seasonal stations [14,15]. In the 1990s, a regular experimental research project commenced within the framework of the international projects "Global Energy and Water Exchange Experiment – Global Asian Monsoon Experiment (GEWEX-GAME)", "Core Research for Evolutional Science and Technology (CREST)" and "Japan Science and Technology (JST)" in cooperation with Japanese and Dutch researchers at the stations of Spasskaya Pad and Neleger. During this project, long-term temperature fluctuations of the upper permafrost, soil moisture, seasonal depth of thaw, water–heat balance and carbon currents were studied [16]. Currently, the monitoring of the ground thermal regime is performed at Chabyda and Yakutsk stations, as well as at several study sites at Ukechi, Umaibyt, Kerdyugen, and along the northern portion of the Tommot–Yakutsk railroad.

The research results of ground thermal evolution within Russia during the period from the Third International Year of Geophysics (1957/59) to the Fourth International Polar Year (2007/08) were evaluated in [17]. Romanovsky et al. [18] assessed the thermal state of permafrost in Russia over the last 20–30 years. Changes in the thermal state of the upper permafrost horizons in Central Yakutia and in natural landscapes over the past 30–40 years have been considered at different stages of research in [19–23].

Since the early 1990s, the unified system of observations of the condition of the geological environment in the area of the perennial and seasonal freezing of the crust of the earth, as well as the assessment, monitoring and forecasting of its changes under the influence of natural, climatic and man-triggered factors, has come to be defined as permafrost monitoring. Permafrost monitoring may be recognized as a method that enables the determination of prospective trends of modern permafrost evolution amidst climate changes and technogenesis [17,24].

The main thermal parameters that may be used as indicators of the thermal evolution of upper permafrost layers in the context of modern climate warming are the depth of seasonal thaw ($\xi$), mean annual temperature at the bottom of the active layer ($T_\xi$), and mean annual temperature at the depth of zero annual amplitude ($T_o$) [17].

In 1981, in order to conduct long-term observations of the thermal state of upper permafrost, the Permafrost Institute organized the Chabyda heat-balance monitoring station 20 km southwest of Yakutsk (Figure 1). The activities of the station and the results of many years of observation are reflected in dozens of publications; additionally, the findings have been referenced in documents published under the auspices of the Federal Service for Hydrometeorology and Environmental Monitoring [25,26]. To date, a significant number of long-term series of observations has been accumulated that have no analogs in Russia and Yakutia. The information obtained at the station allows us to evaluate the response of the upper horizons of permafrost to the climatic fluctuations over recent decades.

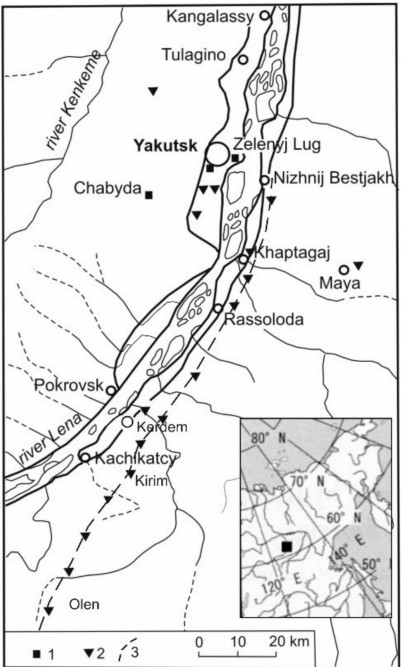

**Figure 1.** Scheme of the work area: 1—stations; 2—polygons; 3—roads and railways.

The major objective of this paper is to evaluate the spatial and temporal variability of the ground thermal state under current climatic changes. To this end, the following goals were realized: a route survey, the selection of objects, the organization of a system of observational networks, the analysis of long-term data from weather stations, regular observations at monitoring sites, and the summarizing and analysis of the obtained data.

## 2. Research Sites and Methods

The Chabyda station is located in the middle taiga zone and is characterized by both low and high-temperature permafrost zones. A detailed description of the observation sites and their location in relation to the station is given in a number of publications [12,13].

The relief of the study area is characterized by a combination of lowered and elevated sections of a hilly–steep plain. Lower areas are the bottoms of streams and gullies, while dry elevated areas are slopes of various steepness and watershed spaces. On the slopes, soils are represented by fine and medium-grained sands, soils on the watersheds in the upper horizons are sandy loam, and those below are sand. The objects of research are soils within the layer of annual heat rotation; in other words, the layer of annual temperature fluctuations (the upper 10–20 m).

Depending on the terrain conditions, soil texture, moisture content, vegetation, and surface cover, the active layer thickness varies over a wide range, from 0.4 to 4.0 m, and the mean annual ground temperature at a depth of 10 m varies from -0.2 to -5.0 °C [13].

Landscape and monitoring methods were utilized to study the ground thermal regime. Landscape studies involved remote sensing and ground surveys of landscapes and their classification and mapping. The information obtained was used for site selection and for the design of a thermal monitoring program.

The experimental sites (S) were organized into two landscape types of terrain. In the stream valley: on the mari (S-1, S-3a); on the grassy lowland (S-8a); in the larch forest (S-8). On slopes: in pine forest on a gentle slope (S-4, S-5, S-6b); in pine (S-7b) and larch (S-9) forests on a watershed slope; in pine forest on moderately steep slopes of the Northern (S-10) and southern (S-11) expositions.

The ground thermal regime within the depth of annual temperature fluctuations is determined both by external (solar radiation, air temperature, and precipitation) and internal (moisture conditions

and lithology) factors. Snow cover, vegetation, and organic mat are also important controlling factors [10–13].

The radiation balance is the result of the incoming and outgoing solar energy at the Earth's surface and is described by the following equation:

$$Q = S + J_{ef} + R \tag{1}$$

where $Q$ is the total incoming radiation; $S$ is the reflected radiation; $J_{ef}$ is the effective radiation (net long-wave radiation); and $R$ is the net radiation.

The components of the heat and moisture exchange of the Earth's surface with the atmosphere are expressed by the heat balance equation or the energy balance equation on the Earth's surface [10]:

$$R = P + LE + B, \tag{2}$$

where $P$ is the sensible heat flux; $LE$ is the latent heat flux ($E$ is the evaporation rate and $L$ is the latent heat of evaporation of water); and $B$ is the ground heat flux.

Long-term studies at a thermal monitoring station in Yakutsk suggest that all components of the energy budget are half or one order of magnitude lower during the cold season compared to the warm season [10]. Differences in the energy balance characteristics among landscape types are therefore more distinctive in summer. Summertime net radiation can be determined from instrumental measurements or by calculation. Pavlov [10] proposed a simple model to estimate the energy balance using incident solar radiation and surface albedo. We measured albedo at the Chabyda research sites during summer months (Table 1). The measured $A$ and $h_o$ were used to calculate the monthly sums of surface net radiation over the warm season, $LE$ and $B$ were derived from instrumental measurements, and P was estimated as a residual of the energy balance equation. Meteorological and actinometric observations were made using standard instruments of Roshydromet.

**Table 1.** Surface albedo at the Chabyda research sites, %.

| Site no. | Surface | Months | | | | |
|:---:|:---:|:---:|:---:|:---:|:---:|:---:|
| | | V | VI | VII | VIII | IX |
| 1 | Shallow lake | 13 | 9 | 9 | 9 | 11 |
| 2 | Moss cover with sedge and reed grass | 18 | 18 | 20 | 20 | 18 |
| 3 | Mosses, ledum and dwarf forest bog | 25 | 15 | 15 | 16 | 15 |
| 4 | Bearberry patches with dead plant spots under a pine canopy | 16 | 17 | 17 | 17 | 19 |
| 5 | Bearberry patches with dominant dead plant spots under a pine canopy | 15 | 15 | 15 | 18 | 18 |
| 6 | Sparse forb mat with dead plant spots at an open site | 16 | 18 | 18 | 21 | 22 |
| 7 | Bearberry patches with dead plant spots under a pine canopy | 14 | 14 | 15 | 16 | 15 |
| 8 | Moss–ledum–lingonberry cover under a birch and larch canopy | 13 | 13 | 12 | 13 | 12 |

The permafrost thermal monitoring program included measurements of ground temperature and thaw depth, as well as the major controlling variables, such as ground material, cryostructure, moisture content, soil unit weight, thermal conductivity of soils and surface covers, and snow depth and density.

Ground temperatures were measured using MMT-4 semiconductor thermistors with an accuracy of 0.1 °C. Until 1990, the measuring devices used were the potentiometer PP-63 and the resistance bridge MO-62; since 1990, the multimeters EDM-169S and EDM-89S have been used. Thermistor cables

placed in backfilled boreholes recorded temperatures at depths of 1, 2, 3, 4, 6, 8 and 10 m. Active layer depths were determined by probing with a steel rod and by hand boring. The analysis and summary of the observation results at various monitoring stages were presented by Skryabin et al. [12] and Varlamov et al. [13].

Observations of the thermal regime of the active layer at the Chabyda station were carried out in the warm seasons of 1981 and 1982 daily, in seven shifts, and in the cold season of 1982–1983 in four shifts. In the annual cycles of 1983–1986, in the summer, observations were carried out once in a pentad, in four shifts. In 1981–1987, soil temperature measurements in the layer of annual temperature variations were carried out once a decade. Since 1987, thermal observations have been carried out according to a more abbreviated program on the 15th of each month. Snow cover is also observed monthly, and at the end of the warm season, the depth of seasonal thawing is observed. The methodology of geothermal monitoring has shown its reliability and can be successfully used in various climatic conditions.

## 3. Results and Discussion

### 3.1. Modern Climate Changes

Since the second half of the 1960s, according to Skachkov [1], Central Yakutia has experienced the highest increasing trend in mean annual air temperature in northern Russia (up to 0.07 °C/year). In 2001–2019, the highest ever temperatures in the history of weather observations were registered in Yakutsk, at −7.7 °C as compared with an average of −10.0 °C. The evaluation of predicted trends of temperature variations in the 21st century by various researchers is ambiguous. For example, predictions of climatic change performed by the Voeikov General Geophysical Observatory (GGO) based on climatic feature extrapolation results suggest that the observed increasing trend in the Russian mean annual air temperature will continue to 2030 by 1.5 °C as compared with values from the year 2000. A further increase of the mean amount of precipitation, predominately in the cold season, is predicted. In the north of eastern Siberia, an increase of precipitation amounts by 7–9% and of snow accumulation by 2–4% in winter is expected [27]. In the late 21st century in eastern Siberia, the increase of the near surface mean annual air temperature is expected to be 6.1 ± 1.6 °C compared to the values of the last two decades of the 20th century [28]. The expected near surface mean annual air temperatures in Yakutsk up to 2015, as estimated at the Permafrost Institute of the Siberian Branch, Russian Academy of Sciences, applying the frequency analysis method, almost coincide with the predictions made by GGO. Furthermore, a temperature decrease of 2–3 °C is expected up to the first half of the 21st century [29,30]. According to predictions made by Neradovky and Skachkov [31], by 2050, an increase of the mean annual air temperature will pass the achieved climatic level of no more than 0.7–1.0 °C. Researchers are increasingly concerned about an expected increase of the amount precipitation in winter and snow accumulation, both of which play a significant role in the increase of soil temperatures.

The variability of the main climate elements (air temperature and precipitation) over the study period at Chabyda station can be determined from the data from Yakutsk weather station—the nearest one to the study area (Figures 2 and 3). We should mention the high correlation between Yakutsk and other weather stations in Central Yakutia [1]. The mean annual air temperature (1961–1990 climate norm) in Yakutsk is -10.0 °C and the mean annual precipitation is 235 mm.

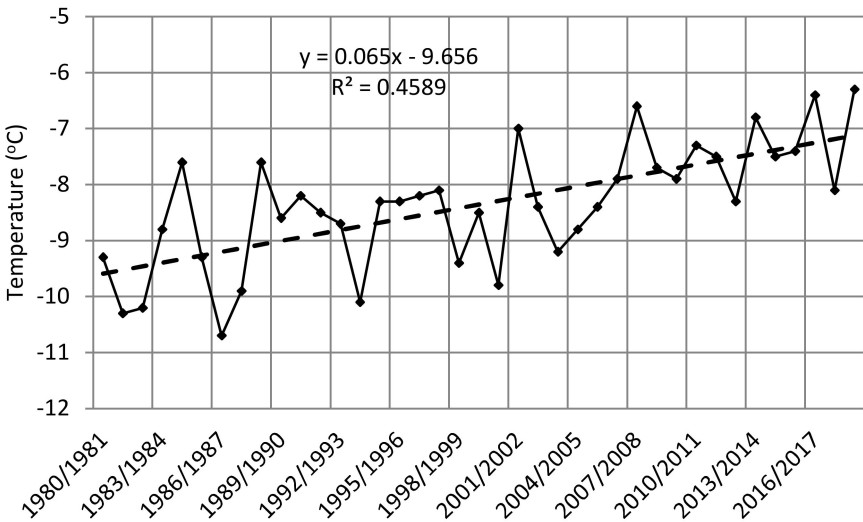

**Figure 2.** Long-term variability (1980–2019) of the mean annual air temperature in Yakutsk (°C). The linear trend is shown by the dotted line.

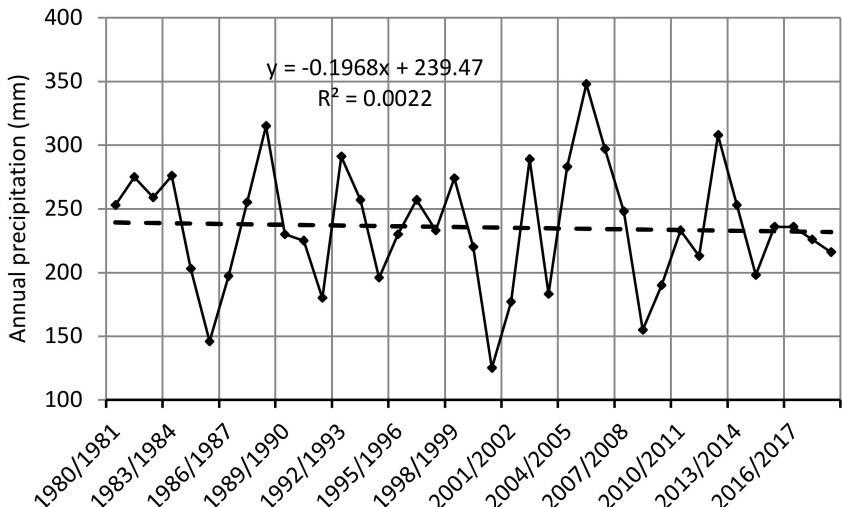

**Figure 3.** Long-term variability (1980–2019) of the annual air precipitation total in Yakutsk (mm). The linear trend is shown by the dotted line.

As shown in Figure 2, the trend of the mean annual air temperature increase is significant and exhibits stable growth. This increase was mainly caused by the growing temperatures in winter (October to April). In summer (May to September), warming was less evident.

The data presented in Figure 3 indicate that the recent decades have seen major year-to-year variations of annual precipitation totals (October to September), but these have not increased in Yakutsk on the whole. The precipitation total by months and for entire years observed during the last 40 years is close to normal. Nevertheless, it should be noted that, in some years, abnormally large (1989,1993,2005–2007,2013) and abnormally small (1986,2001,2004) levels of precipitation occurred.

Thus, the variability of the main climate parameters (air temperature, precipitation, and snow cover) over the past decades is manifested in different ways. The positive trend of the average annual air temperature is the most significant. Precipitation and snow cover height experienced short-period fluctuations without a clear trend.

### 3.2. Surface Energy Balance

A brief review of the radiation regime in the region is presented below based on radiation measurement data from the Yakutsk weather station [12]. The incoming short-wave radiation is largely controlled by sun angle, cloud coverage, and day length. The incident solar radiation varies seasonally over a wide range from 17 (December) to 620–712 MJ/m$^2$ (June). The annual total of radiation is 3710 MJ/m$^2$. Approximately 34% of the incoming radiation is reflected back into the atmosphere. The energy loss is greater during the winter period (75–85%) than in the summer (18–22%). About one-third of the annual incoming radiation is lost by net long-wave radiation. The heat loss by net long-wave radiation increases in proportion to the increase in surface temperature from 29–63 MJ/m$^2$ in the winter months to 167–188 MJ/m$^2$ in the summer. During the winter, the net radiation is negative due to the loss of heat by terrestrial radiation and the reduced solar radiation. Positive net radiation values occur between late March and mid-October. The average annual total of net radiation is 904 MJ/$^2$, or 33% of incoming solar radiation.

The landscape units displayed different energy balance characteristics depending on surficial factors. The partially inundating riparian area covered with mosses and grasses (site 2) showed little variation in albedo during the summer months (18–20%). At the creek valley bottom (site 3), a 0.1-m-thick icing melted out by late May; therefore, the heat loss by reflection was 1.4 times higher than at the riparian site. From June to September, the reflectivity of the surface at site 3 with a moss-shrub cover and hummocky topography was consistently low (15–16%). At the lower and upper slopes (sites 4 and 5), the albedo of bearberry patches with dead plant spots under the open pine canopy varied only slightly over the summer. At the bottom of a drainage trough (site 8), the albedo of the moss-ledum-lingonberry cover under the closed birch and larch canopy was reduced to 12–13 %.

The net radiation ($R$) of the moss-shrub cover on the drainage trough bottom was 1.04 times larger than the $R$ of the riparian site due to the lower albedo. The effective radiation $J_{ef}$ at the open sites comprised similar fractions of $Q_c$ (29%). The reduced solar radiation under the open canopy forest with crown densities of 0.1–0.2 and 0.2–0.3 at the lower slope and the upper slope, respectively, resulted in $R$ being 22–31 % smaller compared to the open sites. The seasonal total of $R$ at the surface under the birch and larch forest with a crown density of 0.7–0.8 was almost twice as low than that at the open site (Table 2).

**Table 2.** Surface energy balance components at the Chabyda sites, MJ/m$^2$.

| Component | Months | | | | | Σ(V-IX) |
|---|---|---|---|---|---|---|
| | V | VI | VII | VIII | IX | |
| **Open areas** | | | | | | |
| $Q$ | 565 | 632 | 603 | 444 | 276 | 2520 |
| **Riparian area (site 2)** | | | | | | |
| $S$ | 102 | 114 | 121 | 89 | 50 | 476 |
| $J_{ef}$ | 165 | 145 | 175 | 134 | 107 | 726 |
| $R$ | 298 | 373 | 307 | 221 | 119 | 1318 |
| $P + LE$ | 238 | 329 | 272 | 205 | 115 | 1159 |
| $B_n$ | 60 | 44 | 35 | 16 | 4 | 159 |

**Table 2.** *Cont.*

| Component | Months | | | | | Σ(V-IX) |
|:---:|:---:|:---:|:---:|:---:|:---:|:---:|
| | **V** | **VI** | **VII** | **VIII** | **IX** | |
| Moss–ledum forest bog (site 3) | | | | | | |
| $S$ | 141 | 95 | 90 | 71 | 41 | 438 |
| $J_{ef}$ | 140 | 151 | 182 | 139 | 102 | 714 |
| $R$ | 282 | 386 | 331 | 234 | 133 | 1365 |
| $P$ | 60 | 227 | 163 | 111 | 83 | 644 |
| $LE$ | 171 | 121 | 133 | 104 | 46 | 575 |
| $B_n$ | 51 | 38 | 35 | 19 | 4 | 147 |
| Lower slope (site 4) | | | | | | |
| $Q$ | 452 | 505 | 482 | 355 | 220 | 2016 |
| $S$ | 72 | 86 | 82 | 60 | 42 | 342 |
| $J_{ef}$ | 141 | 120 | 144 | 111 | 87 | 603 |
| $R$ | 239 | 300 | 256 | 184 | 91 | 1070 |
| $P$ | 159 | 209 | 160 | 122 | 59 | 709 |
| $LE$ | 58 | 67 | 72 | 52 | 28 | 277 |
| $B_n$ | 22 | 24 | 24 | 10 | 4 | 84 |
| Upper slope (site 5) | | | | | | |
| $Q$ | 395 | 442 | 422 | 312 | 191 | 1762 |
| $S$ | 59 | 66 | 63 | 59 | 32 | 279 |
| $J_{ef}$ | 126 | 112 | 133 | 100 | 75 | 546 |
| $R$ | 210 | 264 | 226 | 153 | 84 | 937 |
| $P$ | - | 180 | 138 | 86 | 56 | - |
| $LE$ | - | 59 | 71 | 61 | 30 | - |
| $B_n$ | - | 25 | 17 | 6 | -2 | - |
| Drainage trough bottom (site 8) | | | | | | |
| $Q$ | 266 | 303 | 283 | 210 | 153 | 1213 |
| $S$ | 27 | 31 | 27 | 21 | 16 | 121 |
| $J_{ef}$ | 99 | 91 | 105 | 83 | 53 | 430 |
| $R$ | 140 | 181 | 151 | 105 | 85 | 662 |
| $P+ LE$ | - | 144 | 110 | 74 | 73 | - |
| $B_n$ | - | 37 | 41 | 31 | 12 | - |

The evapotranspiration at each site was determined from measurements with two to four evaporation pans to account for vegetation type (Table 3). Rates of soil water evaporation depend on energy factors and moisture conditions.

**Table 3.** Evaporation from the soil surface at the Chabyda sites, mm.

| Site no. | Surface Cover | Months | | | | | Σ(V-IX) |
|---|---|---|---|---|---|---|---|
| | | V | VI | VII | VIII | IX | |
| 3 | Moss and grass | 78 | 52 | 59 | 38 | 17 | 244 |
| | Moss and ledum | 58 | 45 | 47 | 45 | 19 | 214 |
| 4 | Bearberry | 22 | 26 | 22 | 18 | 8 | 96 |
| | Dead plant spot | 30 | 27 | 34 | 20 | 12 | 123 |
| | Lichen | 12 | 17 | 17 | 13 | 8 | 67 |
| | Lingonberry | 29 | 37 | 43 | 31 | 17 | 157 |
| 5 | Dead plant spot | - | 24 | 28 | 24 | 12 | - |
| 7 | Bearberry | 16 | 15 | 21 | 26 | 10 | 88 |
| | Lichen | - | 17 | 21 | 14 | 14 | - |
| | Dead plant spot | 10 | 41 | 25 | 17 | 8 | 101 |

The evaporation rates on the creek valley bottom were higher between May and July, when the monthly totals of net radiation were largest. In August and September, the reduced net radiation resulted in a decrease in evapotranspiration by a factor of 1.5–2 despite the sufficiently high soil moisture contents. At the forest bog site with a moss-grass-shrub cover, the evapotranspiration averaged over two seasons was 230 mm, which was 1.5 times the mean evapotranspiration from forb meadow at the Yakutsk monitoring station. The seasonal average latent heat flux $LE$ at the forest bog site comprised 42% of $R$.

At the open-canopy forest site on the lower slope (site 4), the evaporative capacity of the vegetation covers was not the same. For example, the evapotranspiration $E_c$ from the lingonberry cover was 2.3 times higher than that from the lichen cover. The evapotranspiration under the pine forest was greatly reduced and averaged 110 mm due to both the reduced $R$ and the lower moisture storage in the sandy soils. Compared to the forest bog (site 3), the evapotranspiration under the pine forest was approximately twice as low. The ratio of $LE$ to $R$ was 0.26. In Central Yakutia, the $LE/R$ values reported by Pavlov [10] were 0.18 and 0.35 for open-canopy pine and larch forests, respectively.

An accumulation of heat in the ground at the riparian site occurred between May and September. The ground heat flux was largest in May and constituted 20% of $R$. In August and September, ground heat fluxes comprised only 3–7% of $R$. Overall, 159 MJ/m$^2$ of heat, or 12% of $R$, entered the ground during the thaw season. In October and November, when the snow cover was shallow, the heat flux was strongly negative, comprising half of the annual heat loss (Figure 4). Between January and April, heat losses were low due to the thick snow cover. The amount of heat utilized in the snowmelt process was 20.9 MJ/m$^2$. Because snow cover depths were greater than the long-term average, the input components of the ground heat flux through the surface exceeded the outputs. The difference between the inputs and outputs comprised 28% of the magnitude of the annual heat exchange.

At the forest bog site, the seasonal sum of sensible and latent heat fluxes, $P + LE$, was 60 MJ/m$^2$ greater than at the bank. Therefore, the ground heat flux was, on a seasonal average, 8% lower despite the higher $R$ values. The $B/R$ was 11%. During the cold season, the heat flux to the atmosphere was strongly reduced due to sustained groundwater flow down the gentle slope onto the bog surface. On an annual basis, the heat input was 1.56 times the heat loss.

The drier sands with a low thermal conductivity under the open canopy (site 4) accumulated 1.7 times less heat during the warm season compared to the forest bog. At site 5, heat accumulation was strongly reduced in the dry sands due to the small temperature gradients near the surface. The $B/R$ ratio was 6%, which was also due to the lower moisture content of the sands.

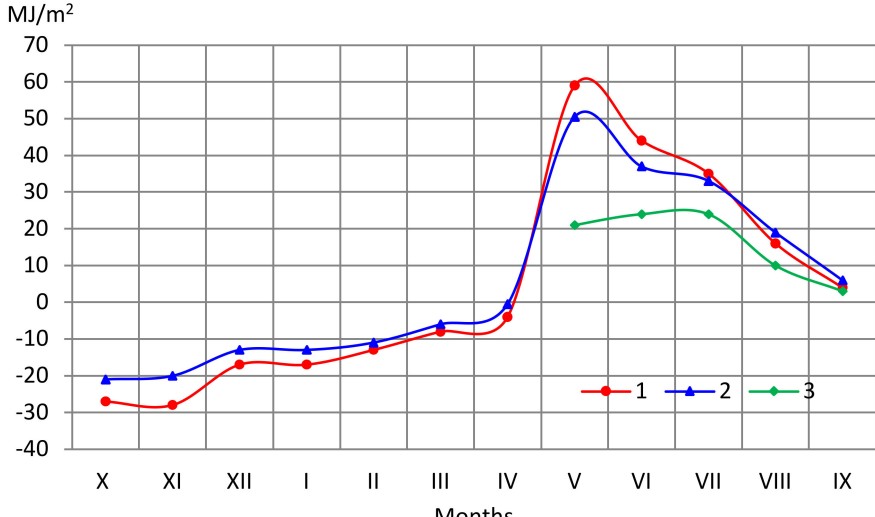

**Figure 4.** Annual course of ground heat flux at the riparian location (1), creek valley bottom (2) and lower slope (3), from Chabyda.

Comparing the absolute values of the energy balance components among the sites, it can be noted that they are of the same order for the gentle slope sites. The only difference is in the magnitude of the sensible heat fluxes, which is attributed to site differences in canopy shading. At the drainage trough bottom (site 8), the *LE* and *B* values are much higher due to the higher soil moisture content and the significant reduction in sensible heat flux under the larch canopy. As a result, the input component of heat exchange at this site is 1.7–1.9 times larger than at the slope sites.

In summary, investigations in the Chabyda area provided important information about the radiation and heat balance characteristics for different landscape units. During the warm season, the albedo is highest for the riparian site with a moss-grass cover (*A* = 18–20%) and lowest for the moss-ledum-lingonberry mat under the larch forest (*A* = 12–13%). The ratio between the seasonal totals of net radiation of open sites and under forest canopies with different crown density varies from 1.25 to 2.0. The totals of latent heat flux vary by a factor of about three among the sites depending on vegetation type and radiation regime. The increased *LE* and *B* values at the drainage trough bottom are attributed to the greater soil moisture content and the reduced sensible heat flux.

### 3.3. Analysis of Long-Term Variability of Soil Temperature

To date, there is a database for sites 1 and 5 for 39 years of observations, for site 8 for 37 years, for site 9 for 35 years, and for other sites there are 34 years of observations. Such representative material allows a qualitative analysis of the variability of the main thermal parameters of the layer of annual temperature variations and allows us to draw objective conclusions. The idea of the long-term variability of quantity $\xi$, $T_\xi$, and $T_{10}$ at the experimental sites is given in Table 4.

The greatest fluctuations of $T_\xi$ and $T_{10}$ occurred in landscapes of a shallow-valley type of terrain (see Table 4 and Figure 5a,b). These changes were mainly determined by the influence of two winter factors: snow accumulation conditions and the sum of air temperatures during the cold period. Moreover, the first factor is prevailing in the sharply continental climate of Central Yakutia [13,32,33].

Over the entire observation period, the lowest and highest average annual temperatures at the sites of the shallow-valley terrain type were noted in hydrological years 2003/04 and 2006/07, respectively (October–September) (see Figure 5). In the slope type of terrain, the peak of the lower temperature of the soil also occurred in 2003/04, and the peak of the temperature increase was noted in 2007/2008. The exception here is site 7b, where the lowest soil temperature was observed in 1987/88 (Figure 6).

**Table 4.** Long-term average and extremes of the main ground thermal parameters within the layer of annual temperature variations.

| Experimental Sites (Observation Periods) | ξ, m | | | $T_\xi$, °C | | | $T_{10}$, °C | | |
|---|---|---|---|---|---|---|---|---|---|
| | Min | Average | Max | Min | Average | Max | Min | Average | Max |
| Small valley | | | | | | | | | |
| 1 (1981–2019) | 0.81 | 1.06 | 1.30 | −5.1 | −2.8 | −0.6 | −3.4 | −2.7 | −1.8 |
| 3a (1986–2019) | 0.37 | 0.46 | 0.53 | −7.4 | −5.0 | −1.3 | −4.9 | −3.9 | −2.8 |
| 8 (1982–2019) | 0.86 | 1.17 | 1.37 | −5.5 | −3.6 | −1.3 | −3.3 | −2.7 | −1.9 |
| 8a (1986–2019) | 0.65 | 1.02 | 1.45 | −6.5 | −3.3 | 0.1 | −4.5 | −3.3 | −1.8 |
| Slope | | | | | | | | | |
| 5 (1981–2019) | 3.26 | 3.50 | 3.86 | −0.1 | −0.4 | −0.1 | −0.6 | −0.4 | −0.3 |
| 6б (1986–2019) | 3.54 | 3.78 | 4.11 | 0.0 | −0.4 | 0.0 | −0.6 | −0.4 | −0.2 |
| 7б (1986–2019) | 1.70 | 1.87 | 2.00 | −0.4 | −1.2 | −0.4 | −1.5 | −0.9 | −0.3 |
| 9 (1985–2019) | 1.31 | 1.51 | 1.72 | −1.0 | −2.5 | −1.0 | −2.5 | −2.2 | −1.8 |
| 10 (1986–2019) | 1.63 | 1.91 | 2.1 | −0.7 | −1.8 | −0.7 | −2.4 | −2.0 | −1.6 |
| 11 (1986–2019) | 1.73 | 1.91 | 2.27 | −0.2 | −1.0 | −0.2 | −1.5 | −1.2 | −0.9 |

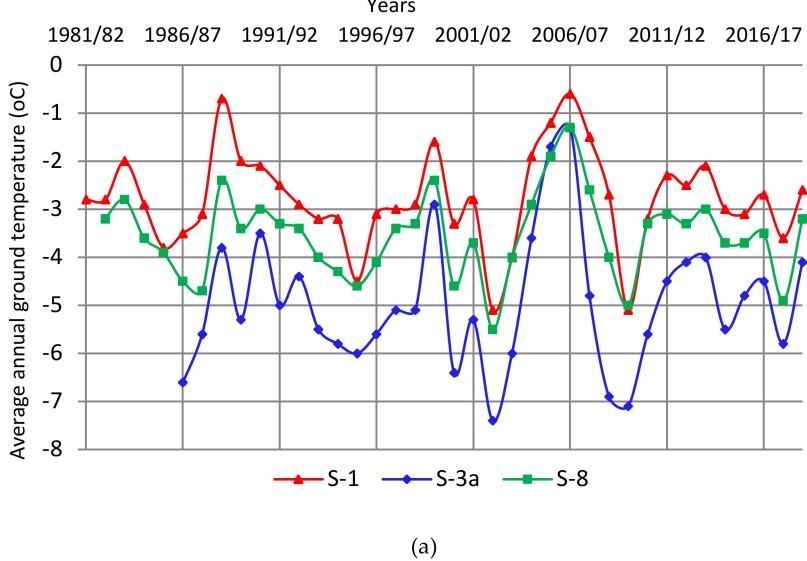

(a)

**Figure 5.** *Cont*.

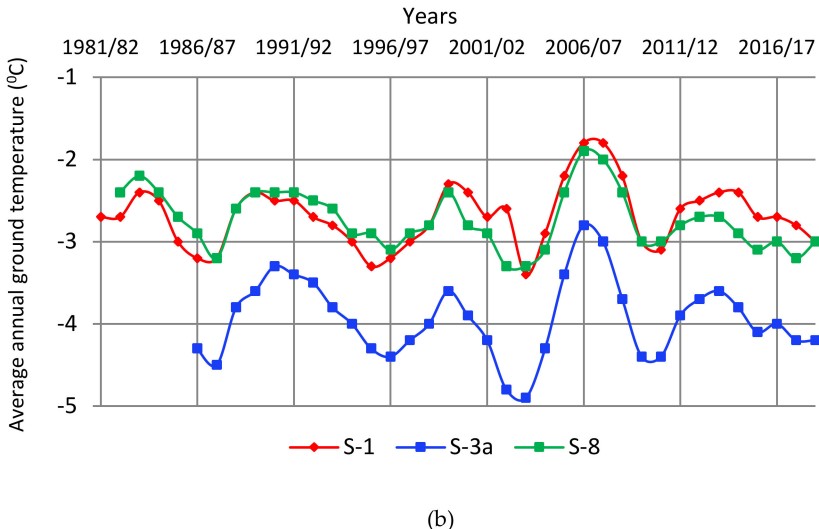

(b)

**Figure 5.** Long-term variability of ground temperatures at the base of the active layer (**a**) and at a depth of 10 m (**b**) in the shallow-valley type of terrain at the Chabyda station.

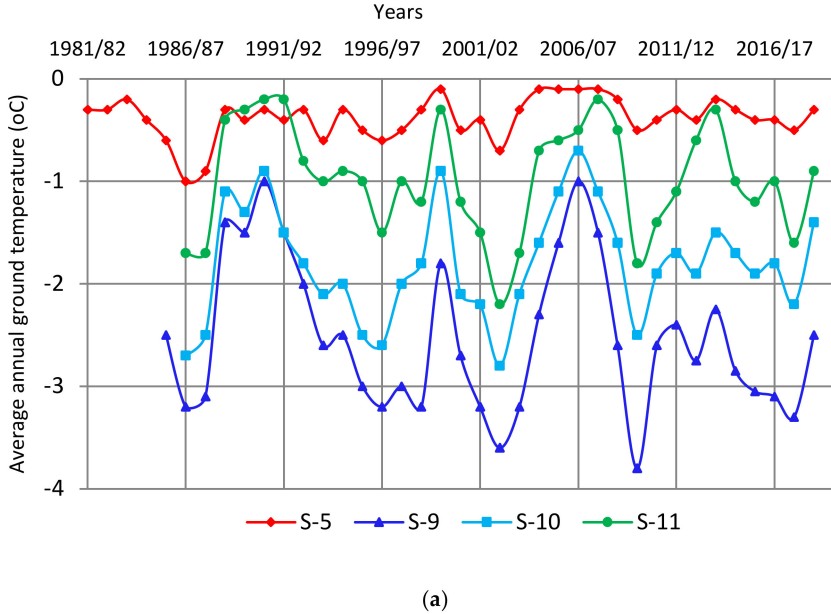

(**a**)

**Figure 6.** *Cont.*

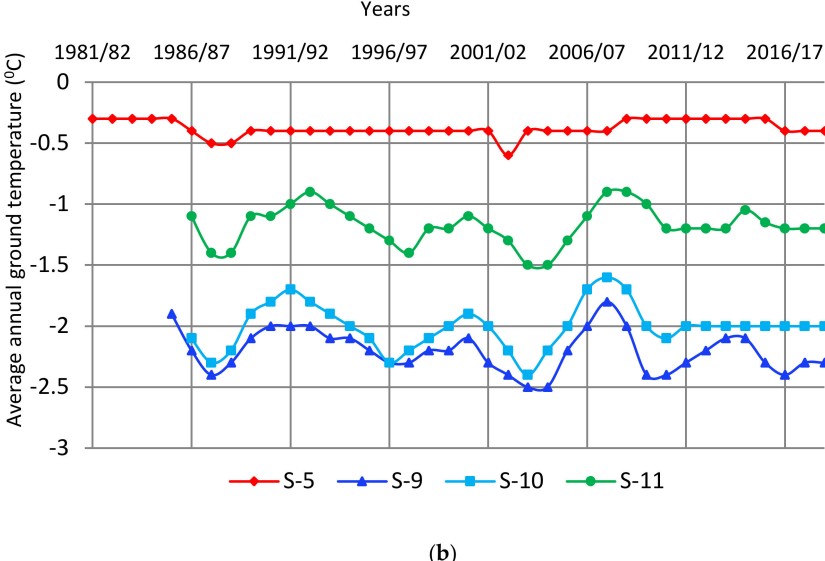

(**b**)

**Figure 6.** Long-term variability of ground temperatures at the base of the active layer (**a**) and at a depth of 10 m (**b**) in sthe lope type of terrain at the Chabyda station.

The winter in 2002/03 featured abnormally light snow and had an abnormally late period of formation of stable snow cover (Figure 7). This was the main reason for the strong cooling of the soil, despite a fairly warm year. Subsequent years were characterized by snowy winters and higher than normal rainfall, which led to a sharp increase in soil temperature. In the period from 2002/03 to 2006/07 (2007/08), the temperature of the soils at the bottom of the active layer in these types of terrain increased by 0.5–6.0 °C and at the depth of zero annual amplitude by 0.3–2.7 °C (see Figures 5 and 6).

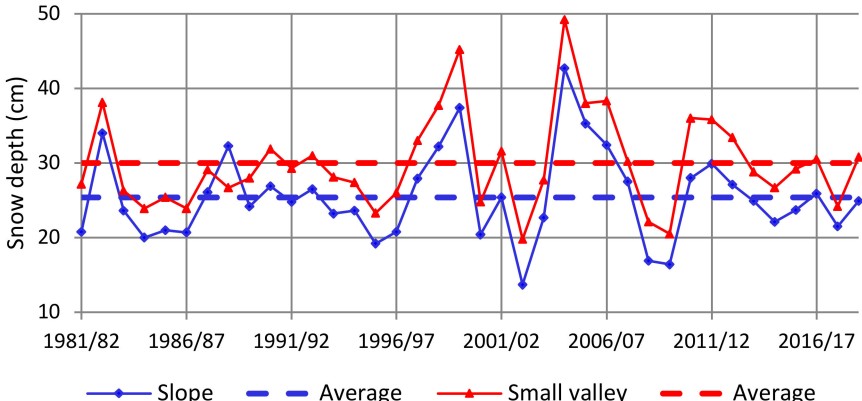

**Figure 7.** Long-term variability of the mean winter snow depth in the slope and shallow-valley types of terrain at the Chabyda station (dashed and dotted lines are long-term average values).

The warming of the permafrost and deepening of the active layer have accelerated thermokarst, thermal erosion and other permafrost processes in the region. Water ponding and paludification have been observed in the low lying areas. These changes have resulted in increased water levels in the existing lakes, the filling of the previously dry lake basins, and in the rise of the water table in the active layer. In the water-logged topographic lows, residual thaw layers have developed due to the lowering of the permafrost table and significant warming of the near-surface ground [20,23]. Thus, 2007 and 2008 were critical years for the permafrost thermal regime in the 40-year period of observations or even in its century-long history.

The winters of 2008/09 and 2009/10 were characterized as abnormally snowy with late periods of formation of stable snow cover; thus, the soils were much cooled to almost the level of 2002/03, despite the abnormally warm years.

Subsequent winters were characterized by close to long-term average values of average winter snow depths, and therefore, mean annual ground temperatures at the active-layer bottom and at the depth of zero annual amplitude were close to long-term average values. It should be noted that the correlation between the temperatures at the bottom of the active layer and the average winter snow depths is quite high (correlation coefficients are 0.55–0.74) [13].

According to long-term observations, trends in the increasing average annual temperatures of the layer of annual heat rotations are extremely weakly expressed. Noticeable trends in temperature increase were observed only at two sites: 7b (0.23°C/10 years) and 8a (0.33°C/10 years) (Table 5). This was largely caused by the abnormally snowy winters mentioned above, and site 8a was flooded during several warm seasons with meltwater. In addition, the first years of the analyzed period (1986–1988) were quite cold. It should also be noted that in these types of terrain at sites 8 and 9, there are decreasing trends of $T_{10}$. This is due, first of all, to an increase in the shading of the surface under the canopy of the stand with its crowns and the growth of shrubs and shrubs.

**Table 5.** Trends in mean annual ground temperature at the depth of zero annual amplitude (10 m).

| Terrain Type | Ground, Experimental Site (S) and Number | Observation Periods | Trends (°C/10 year) |
|---|---|---|---|
| Small valley | Sand (S-8, S-1) | 1981–2019 | −0.1–0.05 |
| | Turf, sand (S-3a, S-8a) | 1986–2019 | −0.02–0.33 |
| Slope | Sand (S-5, S-6b, S-7b) | 1981–2019 | ~0.00–0.23 |
| | Sandy silt, sand (S-10, S-11) | 1986–2019 | ~0.00–0.02 |
| | Loam, sandy silt, sand (S-9) | 1985–2019 | −0.05 |

According to long-term observations, the thermal effect of abnormal winters has been experimentally established. Thus, one warm and less snowy winter can lower the temperature of the soil more than a cold and snowy one. An abnormally cold winter can produce a stronger effect than several abnormally warm ones following one after another [32].

The most important conclusion from the results of long-term observations is that the temperature of the soils experiencing significant interannual and short-period fluctuations associated mainly with the great variability of the snow accumulation regime has a very weak tendency to increase. Against the background of significant changes in the average annual air temperature (in Yakutsk for the period 1981–2019, the growth trend was 0.064 °C/year), the thermal state of the layer of annual temperature variations at the slope-type sites of the Chabyda station area remains stable.

*3.4. Analysis of the Long-Term Variability of the Depth of Seasonal Thawing*

As is known, the depth of seasonal thawing (ξ) depends on the mechanical composition of soils, their moisture content, the nature of the vegetation and soil cover. The greatest long-term variability Δξ is typical for soils of a shallow-valley type of terrain and makes up 30–55 % of their maximum extremum. In the slope type of terrain, Δξ varies within the range of 14–24 % (see Table 4 and Figure 8). It was previously believed that the main factors determining the long-term variability of the thickness of the seasonally thawed layer are the sum of positive air temperatures and summer precipitation. However, recent developments on this issue show that long-term changes in the depth of seasonal thawing and the sums of summer air temperatures do not correlate sufficiently well [34].

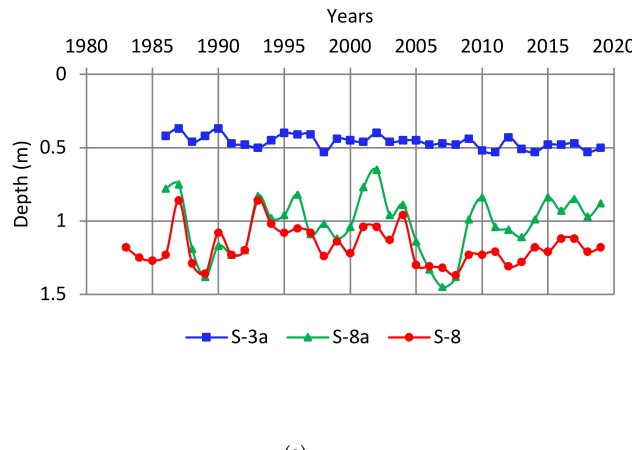

(a)

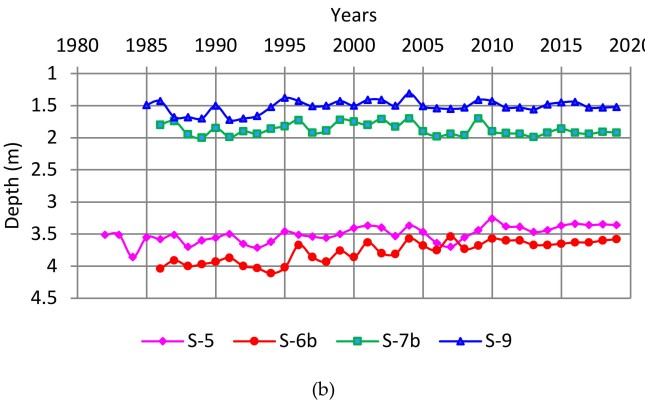

(b)

**Figure 8.** Long-term variability of the active-layer thickness in the shallow (**a**) and slope (**b**) types of localities at the Chabyda station.

The highest values of ξ (3.54–4.11 m) were noted at site 6b (the upper part of the gentle sandy slope), with the smallest (0.37–0.53 m) noted at Site 3a (the bottom of the brook valley with peat soil). At ten sites of the station, on average, the maximum thickness ξ (1.81 m) occurred in 1988–1989 and 2006–2008.

Most likely, the ground conditions—namely the moisture content—can majorly affect the thawing depth. An analysis of the data obtained at the station shows that the largest long-term variabilities of ξ are characteristic of landscapes of a shallow-valley type of terrain (up to 80 cm). On sites in the slope type of terrain, these changes were 30–60 cm (see Table 4).

In the shallow-valley type of terrain at sites 3a and 8, a significant increase in the depth of seasonal thawing was noted. Significant decreasing tendencies are observed at sites of the slope type of terrain (sites 5, 6b, 9, 11). Additionally, if in the watershed area (Site 9) this decrease can be attributed to the intensive growth of shrubs, then for sites 5 and 6b, fluctuations in the level of permafrost waters of the seasonally thawed layer most likely play a significant role. At other sites (7b, 10), with significant interannual changes in the depth of seasonal thawing, insignificant growth tendencies are observed (Table 6).

**Table 6.** Trends in the depth of seasonal thawing of soils of the Chabyda station.

| Terrain Types | Ground, Experimental Site (S) and Number | Observation Periods | Trends (sm/10 Year) |
|---|---|---|---|
| Small valley | Turf (S-3a) | 1986–2019 | 2.4 |
| | Sand (S-1, S-8) | 1981–2019 | −0.8–2.4 |
| | Turf, sand (S-8a) | 1986–2019 | −2.2 |
| Slope | Sand (S-5, 6b) | 1981–2019, 1986–2019 | −7.1−−13.9 |
| | Sand, sandy silt (S-7b) | 1986–2019 | 2.2 |
| | Loam, sandy silt (S-9) | 1985–2019 | −3.0 |
| | Sandy silt, sand (S-10, S-11) | 1986–2019 | 2.2−−3.0 |

The maximum depths of seasonal thawing at individual sites were observed in different years. In the shallow-valley type of terrain (sites 1, 3a, 8, 8a), they were noted in 2007–2008. In the slope type of terrain, the maxima $\xi$ were revealed in the following years: site 5—1984, site 6b—1995, site 7b—1989, site 9—1991, and site 10 and 11—1988.

Thus, during climate warming, even within local undisturbed areas, an opposite trend can be detected in modern changes at the depth of the seasonal thawing of soils. This serves as strong evidence that the thickness of the active layer is not a sensitive indicator of modern climate change.

The analysis of a large amount of experimental data confirmed the conclusion about the weak response of the depth of seasonal thawing to modern climate changes [35]. The observations convincingly show that under the conditions of modern climate warming, an increase in the depth of seasonal thawing of soils does not always occur. Maxima and minima of this value at various sites of the station are observed, most often in different years. This indicates that the depth of seasonal thawing depends not only on long-term changes in air temperature in the warm period but also on other meteorological factors.

## 4. Conclusions

At the end of the last century, we concluded that the thermal regime of permafrost soils was stable and their reaction to climate variability was weak [36]. Subsequent years have confirmed this conclusion.

Based on the analysis of the 39-year monitoring, the main conclusions are as follows:

1. The radiation and heat balance characteristics have been determined for different landscape conditions in the Chabyda area.
2. The long-term dynamics of the thermal state of the layer of annual temperature variations during climate warming indicates the thermal stability of both high-temperature and low-temperature permafrost. The main regulatory factor in the dynamics of the thermal state of permafrost soils is the snow accumulation mode.
3. The depth of seasonal thawing, despite significant interannual fluctuations, weakly responds to climate warming and has no significant trends.
4. The results of studies of the thermal regime of soils can be extended to similar landscapes of Central Yakutia.
5. The practical value of the materials obtained at the Chabyda station also lies in the fact that the observation results can be used to model heat transfer processes in natural landscapes.

**Author Contributions:** All authors—S.P.V., Y.B.S., and P.N.S.— made a significant contribution to the published work. All authors have read and agreed to the published version of the manuscript.

**Funding:** This research received no external funding.

**Acknowledgments:** Support for the long-term thermal monitoring of upper permafrost has been provided by the Melnikov Permafrost Institute Siberian Branch of the Russian Academy of Sciences as part of its fundamental research programs, agreements with the Institute "Mosgiprotran" of the USSR, JSC "Projecttransstroy" of the Russian Federation, the Ministry of Transport of the Republic of Sakha (Yakutia), JSC "Railways of Yakutia". The manuscript was prepared with the financial support of RFBR grants 18-44-140008 and 18-45-140029.

**Conflicts of Interest:** The authors declare no conflict of interest. The funders had no role in the design of the study; in the collection, analyses, or interpretation of data; in the writing of the manuscript, or in the decision to publish the results.

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
