# Peer review of "Influence of Climate Change on the Thermal Condition of Yakutia’s Permafrost Landscapes (Chabyda Station)"

_land, doi:10.3390/land9050132_

Round 1
Reviewer 1 Report
Please find the attachments for the comments and suggestions.

Author Response
Responses to reviewer's comments 1
Review of “Influence of climate change on the Thermal condition of Yakutia’s permafrost landscapes (Chabyda Station)”
A brief summary
This study aimed to present the thermal state of permafrost landscapes under changing climate in the Yakutia region, Russia. Authors have presented several kinds of dataset including air temperature, precipitation, soil temperature at different depth and analysis of ground materials at different sites of two types of terrain. Overall, this study presented the long-term thermal regimes of the permafrost region in the Russia.
First of all, thank you very much for reviewing our paper; your advice, comments, and suggestions are helpful and constructive. We have carefully modified our manuscript based on your review.
Broad comments
This paper presented the thermal state of the permafrost landscapes using the long-term measurements at the Chabyda station. This study is largely supported by the vast amount of dataset and author have presented continuous and long-term data for the study sites. However, this paper lacks well structured frame. Study sites and methods need to be clearer, which should be easy to understand. The study site (Area map) is not presented with the experimental sites. Although author have mentioned that study site can be seen in the past studies, however, past studies are in local language make difficult to follow. If this study have new findings and applied nobel approach, if must have nice topographical map, otherwise it is difficult to follow the manuscript. A map of the research area was added to the manuscript (figure 1).
I suggest integrating all the methods including methods for energy balance together in the section methods. It's done.
Similarly, manuscript explained about the data, their trends, however, it lacks the discussion. Therefore, I suggest adding the discussion for the each results. Also, please clearly mention the new contribution of this research in science, community.
Introduction of the manuscript failed to include past similar research and no clear motivation of this manuscript is noticed and several similar study are found already published in the study region. Manuscript is failed to elaborate the spatial variability of the ground thermal state based on the major objective. The quality of figures also need to be improved. Some of the data can be presented as supplementary file and figure rather than the large table. I really wish and hope that authors can improve the manuscript for the publication.
Specific comments
L – Refers to a line number
Title
Abstract
Abstract is well written
Introduction
The introduction is well written with nice flows of sentences, which are all connected to one another. However, I would like to see some clarity about research frame. It seems to be existence of several past studies in the Central Yakutia and several of them are published in the Russian language. The introduction part should introduce the lacking part in the past studies and motivation of this study.
Research motivation is reflected in lines 27-34.
Pleased add this reference, i.e., Varlamov et al., 2019 Evolution of the thermal state of permafrost under climate warming in Central Yakutia.
Yes, added.
L35-38: Are you talking about the results from the same region?
Yes. This is the district of Yakutsk and the territory of the Leno-Amginsky interfluve.
L44: Mentioning the name of the person is necessary in context of the supervision, please rewrite the sentence by providing the reference only.
Yes, it's done (see lines 41-43).
L55-57: Please rewrite the sentence “Based on that research,…..….” It seems like “During this project” rather than “based on that research”.
A correction was made (see lines 54-56).
L63-65: What about your study site, is it a same like yours “central Yakutia”? I t seems like study has been already done to understand the thermal state of permafrost. Please make it clear if there is anything else.
In relation to Central Yakutia, the territory of the Chabyda station is one point. On the territory of this station, the permafrost conditions change within a large range (see lines 100-102), which fully reflects the variability of permafrost conditions in the region.
L64: “….permafrost layers in the Central Yakutia,…...
A correction was made (see lines 61-63).
L72-75: Please provide the references.
A link was made (see line 74).
L80-82: What about the study that is mentioned in line 63 to 65?
A correction was made (see line 78).
2 Research Sites and Methods
I recommend keeping the study site showing the all the experimental plots in the study site. Past studies in Russian language are difficult to follow. The spatial variability of the ground thermal state of the regions is difficult to follow (major objective of this research) in the manuscript without adequate information about different sites, their distribution, elevation and other.
L115-116: Please check the space between two lines.
Yes.
- Results and discussion
3.1 Modern climate changes
L142: …….increasing trend in Russian mean annual air temperature???
A correction was made (see lines 168-171).
L146: …..by 2-4%?
A correction was made (see line 173).
L146-147: please check the space between two lines.
Yes.
L151: Furthermore, a temperate decrease of 2-3 degree Celsius is expected….., how it is predicted?
This is a question for the authors we refer to.
L161: ……in the Yakutsk is -10.0 °C and the ……..
A correction was made (see lines 187-188).
Figures: Please improve the quality of the figure; especially font does not look better.
Yes.
L168-170: Is the increase of temperature is statistically significant. I suggest conducting the statistical test of the trend and for the precipitation too.
Yes, it is checked. It's true.
3.2 Surface energy balance
Energy balance components are estimated for the different sites and comparison of the component has been done nicely. However, it lacks the discussion of the energy balance components. Why components are different from each other, and what will be effect of this on permafrost? This kind of analysis seems lacking.
The authors agree with the reviewer. The components of the power balance of stations, depending on their permafrost-landscape conditions, differ from each other. We will take into account the reviewer's wishes in our future works.
L177-187: This part need to move in methodology section rather than in the results. These equations seems to be already published in several publications, which need to citated. These lines are moved to section 2 (see lines 113-127).
L205: What is the period for the measurement of albedo, and what V VI VII VIII and IX indicate in the table?
Tables 1-3 have been reworked. Added the "months" line.
I think table of the albedo seems unnecessary, better put as supplementary file.
Table 1 has been moved to section 2.
What is LE и in line 207?
LE is the heat expenditure for evaporation (E is the amount of evaporation; L is the heat of evaporation of water).
L217: What do you mean by (15 16 %)?
15-16 % is the surface albedo at station 3 in June-September (see table 1).
I recommend keeping the figure 4a and 5a in one figure. Similarly, figure 4b and 5b can be keep together. Also, please find out the trend of the ground temperature. It is difficult to see the increasing temperature from the figure 4a, b and 5a and b. it is difficult to say warming of the permafrost have accelerated thermoskarst, thermal erosion and other processes based on the temperature data presented here. Please provide clear evidence.
Due to the visual load, figures 4 and 5 do not show the trend lines of soil temperature changes. They are given in the table 5 in quantitative terms. Figures 4 and 5 were reworked as recommended by the reviewer. Warm winters, wet summers and snowy winters in 2005/06-2007/08 contributed to an increase in the average annual soil temperature and an increase in the power of the active layer, which accelerated the development of thermokarst processes, thermal erosion and flooding of negative landforms. This was previously published by the authors in papers 20 and 33.
- Conclusion
L407-409: It is difficult to follow the sentence, please rewrite the sentences.
Yes (see lines 412-414).
Reviewer 2 Report
General comments:
This study presents a 39-yr field observation to investigate the influence of climate change on thermal situations of Siberian permafrost. After reading this manuscript, I feel that the study is well-written and has significant contributions to understand the pan-arctic permafrost under the context of climate change. I suggest accepting the paper after minor revisions in MDPI Land.
Specific comments:
Line 32-44: Introduction is poorly organized and difficult to read. There are no clear historical observations on dynamic permafrost thermal conditions, nor highly summarize the important results of those previous studies. It is important to propose your scientific questions instead of focusing on analyzing time-series data.
Line 90: What’s the exact location?
Line 137: oC. Symbol issue?
Line 157: Should put in methodology
Line177-187: Should put in methodology
Line210: Table 1 should move to the Support information. Too many tables in this section and not each of them is informative in the context.
Line 301, 307, 374: Captions of S-5, B-1 in Figure4/5/7 are hard to be understood. Please describe with details either in the main text or in the caption. Otherwise, name the station 5,8…directly using their locations throughout the manuscript.
Line336: Years in the x-axis are too dense in Figure 6.
Line 406: The conclusion is over subjective. What’s the significance of the study? What are major contributions according to your own data and findings?
Line 409: What thesis? Be informative.
Author Response
Responses to reviewer's comments 2
General comments:
This study presents a 39-yr field observation to investigate the influence of climate change on thermal situations of Siberian permafrost. After reading this manuscript, I feel that the study is well-written and has significant contributions to understand the pan-arctic permafrost under the context of climate change. I suggest accepting the paper after minor revisions in MDPI Land.
Thank you very much for the positive evaluations on our work. We have carefully modified our manuscript as the below comments.
Specific comments:
Line 32-44: Introduction is poorly organized and difficult to read. There are no clear historical observations on dynamic permafrost thermal conditions, nor highly summarize the important results of those previous studies. It is important to propose your scientific questions instead of focusing on analyzing time-series data. Reviewer 1 noted that the introduction was well written, but gave 7 comments that were excluded (see lines: 41-43, 54-56, 61-63, 74, 78, 80-81).
Line 90: What’s the exact location? The location of the Chabyda station is given on lines 77, 88-89.
Line 137: oC. Symbol issue? The release symbol was corrected to 0.07 oC/year (see line 165).
Line 157: Should put in methodology. The authors wish to leave it as it is, so as not to disrupt the logic of the text.
Line177-187: Should put in methodology. The authors agree with the reviewer and put in section 2 (lines see lines 117-127)
Line210: Table 1 should move to the Support information. Too many tables in this section and not each of them is informative in the context. The authors moved table 1 along with the text (lines 200-211) to section 2, following the reviewer's comment 3 (see lines 128-140).
Line 301, 307, 374: Captions of S-5, B-1 in Figure4/5/7 are hard to be understood. Please describe with details either in the main text or in the caption. Otherwise, name the station 5,8…directly using their locations throughout the manuscript. In figures 4A and 5A, the designation B-1 is corrected to S-1. The description of the stations is given in section 2 (lines 107-112), so it makes no sense to give detailed descriptions of them in the caption of the drawings.
Line336: Years in the x-axis are too dense in Figure 6. Years on the x axis have been discharged.
Line 406: The conclusion is over subjective. What’s the significance of the study? What are major contributions according to your own data and findings? In conclusion, the main conclusions obtained from the results of almost 40 years of field observations conducted by the authors of the manuscript of the article are presented. On the contrary, we consider these conclusions to be objective.
Line 409: What thesis? Be informative. Subsequent years have confirmed this concluded.

Reviewer 3 Report
The manuscript is very interesting and provides a detailed analysis of a very long sequence of data. The results seems to be quite consistant and interesting. However, i am mainly worry about the description of the data and methods. From my point of view, they are not described in detail in the methods section. Sometimes the methods are presented in the results section, and in other cases, they are not presented at all. Moreover, many important results are based on analyses that are not described in the manuscript or in statistics neither provided on tables or plots. So, the reader is not able to check what the authors say about the results.
Moreover, it seems the authors you only some part of the data... If i am right, the authors have data from boreholes located in most of th study sites. Why they do not use these data for more things than just only the calculation of the temperature of the Zero-thermal Amplitude? May be there is a good reason. If explained, it could be valid.
The detailed comment are these:
Lines 89-128: It miss a description of the instrumentation, may be using a table. In the introduction, a nice and detailed description of the evolution on the monitoring sites and instrumentation is provided, but it should be also summarized in a table or described in detail. For example, it was not clear in the introduction the existence of radiation instruments but In the results section, these data are used to derive an energy balance.
Lines 130-133: It is right, but I recommend adding some references to support this idea. Everybody knows, but it is better to provide some references.
Lines 177-189: This should me moved to the section 2, to have in the same place all the data and methods description.
Lines 200-211: All this information should go to section 2 again. This is a description of the method, but the authors present it in the results section.
Lines 292-293: How do you calculate the long-term variability? This is not explained in the methods section. Describe the methods and provide the references. How do you calculate the values of Table 4? By the way, what is T0? The temperature of the zero-thermal amplitude? Clarify, because this acronym was not used before in the manuscript. What is about the other parameters at the head of the table?
Lines 298-299: How do you know that? Where are the evidences? You do not show a plot of the snow thickness evolution or a calculation of the correlation among these parameters. So, how do you know it? It is true that you show latter a plot of the long-term variability on the snow cover, but you do not describe here the correlations among both parameters.
Lines 301-310: The Y axis of Figures 4 and 5 is “annual ground temperature” What temperature? Mean? Maximum? Minimum? Clarify.
Lines 340-342: Where can I see these results of the correlation? I do not see them anywhere in the manuscript… moreover this correlation was not described in the methods section.
Lines 344-345: Again, I do not see the results of the statistics, so I do not see where the results of the correlation are statistically significant or not.
Lines 351-352: I do not know how to interpret the 3rd and 5th columns. By the way, what tis the Thermal Type? How do you define it? Where this classification comes from?
Lines 366-367: Again, how do you calculate the long-term interannual variability
Line 365: Here you talk about the seasonal thawing. If I understand correctly based on your description at the beginning of the manuscript, you measured it by mechanical probing once a year. This is right… however in the figure 7 you talk about active-layer thickness… Then, are you talking about active-layer, calculated based on the thermal data? Or thaw depth measured by mechanical probing, that could be or not coincident with the active layer depending when do you measure it in the process of thawing of the ground? If I understooth correctly, the study sites have boreholes, right? Why not to calculate the real active layer thickness by the temperatures?
Line 381: interannual variability or long-term variability?
Line 384: Again, where can I see the results of the statistics?
Lines 390-391: I do not know how to interpret the last column
In summary, the mauscript and the science inside is very interesting and the data seems to be impresive, but i think that the manuscript need small improvements to help the reader to follow all the ideas presented inside, and the final conclusions.
It is a very interesing manuscript that i would like to see publish after its minor revision. Well done!
Author Response
Responses to reviewer's comments 3
The manuscript is very interesting and provides a detailed analysis of a very long sequence of data. The results seems to be quite consistant and interesting.
First of all, thank you very much for reviewing our paper; your advice, comments, and suggestions are helpful and constructive. We have carefully modified our manuscript based on your review.
However, i am mainly worry about the description of the data and methods. From my point of view, they are not described in detail in the methods section. Sometimes the methods are presented in the results section, and in other cases, they are not presented at all. Moreover, many important results are based on analyses that are not described in the manuscript or in statistics neither provided on tables or plots. So, the reader is not able to check what the authors say about the results.
The authors agree with the reviewer's comment that the methods are presented in different sections. In the presented version of the manuscript, the authors tried to take this remark into account.
Moreover, it seems the authors you only some part of the data... If i am right, the authors have data from boreholes located in most of th study sites. Why they do not use these data for more things than just only the calculation of the temperature of the Zero-thermal Amplitude? May be there is a good reason. If explained, it could be valid.
30-40 years ago, the world scientific community was not faced with the problem of climate warming in the permafrost zone. In the late 80-ies of the last century, due to climate warming, scientists began to study the reaction of permafrost to this warming. The authors started studying the thermal regime of soils at the Chabyda station in 1981. Detailed studies of the hydrothermal regime of soils were carried out before 1986. Since 1987, the study of the temperature regime of soils is carried out under a reduced program and continues to this day. In this paper, the thermal regime of soils is given according to field observations, and not calculated. According to the authors, the value of research lies in this.
The detailed comment are these:
Lines 89-128: It miss a description of the instrumentation, may be using a table. In the introduction, a nice and detailed description of the evolution on the monitoring sites and instrumentation is provided, but it should be also summarized in a table or described in detail. For example, it was not clear in the introduction the existence of radiation instruments but In the results section, these data are used to derive an energy balance.
The authors agree with the reviewer's comment. In the presented version of the manuscript of the article, the description of control and measuring devices for measuring resistance is given (see lines 146-147). Meteorological and actinometric observations were made using standard instruments of Roshydromet (see lines 137-138).
Lines 130-133: It is right, but I recommend adding some references to support this idea. Everybody knows, but it is better to provide some references.
This paragraph was supplemented with several links (see lines 113-116).
Lines 177-189: This should me moved to the section 2, to have in the same place all the data and methods description.
On the reviewer's recommendation, lines 177-189 were moved to section 2 (see lines 117-127).
Lines 200-211: All this information should go to section 2 again. This is a description of the method, but the authors present it in the results section.
On the reviewer's recommendation, lines 200-211 were moved to section 2 (lines 128-140).
Lines 292-293: How do you calculate the long-term variability? This is not explained in the methods section. Describe the methods and provide the references. How do you calculate the values of Table 4? By the way, what is T0? The temperature of the zero-thermal amplitude? Clarify, because this acronym was not used before in the manuscript. What is about the other parameters at the head of the table?
The minimum and maximum average annual and average multi-year annual ground temperature values give an estimate of the long-term variability of ground temperature at each station (see table 4). To make it clear, the abbreviation Was converted to T10, which means the average annual soil temperatures at a depth of 10 m. There is a link to this abbreviation in the manuscript of the article.
Lines 298-299: How do you know that? Where are the evidences? You do not show a plot of the snow thickness evolution or a calculation of the correlation among these parameters. So, how do you know it? It is true that you show latter a plot of the long-term variability on the snow cover, but you do not describe here the correlations among both parameters.
Here, due to the organic nature of the article, the authors limited themselves to references to previous works by Yu.B. Skachkov (31, 32) and a joint monograph (13), where the correlation between the average winter snow thickness and the average annual soil temperature is considered in detail.
Lines 301-310: The Y axis of Figures 4 and 5 is “annual ground temperature” What temperature? Mean? Maximum? Minimum? Clarify.
This is the average annual soil temperature. Corrected the Y-axis captions in figures 5 and 6.
Lines 340-342: Where can I see these results of the correlation? I do not see them anywhere in the manuscript… moreover this correlation was not described in the methods section.
A reference was made to an earlier work. Correlation between parameters is a separate topic that is beyond the scope of this article. This is why it was not described in the methods section (see line 347).
Lines 344-345: Again, I do not see the results of the statistics, so I do not see where the results of the correlation are statistically significant or not.
The proposal has been edited.
Lines 351-352: I do not know how to interpret the 3rd and 5th columns. By the way, what tis the Thermal Type? How do you define it? Where this classification comes from?
To avoid discrepancies, the authors removed column 3 in table 5. Thermal characteristics of soils in table 4.
Lines 366-367: Again, how do you calculate the long-term interannual variability
∆ξ = ξmax – ξmin.
Line 365: Here you talk about the seasonal thawing. If I understand correctly based on your description at the beginning of the manuscript, you measured it by mechanical probing once a year. This is right… however in the figure 7 you talk about active-layer thickness… Then, are you talking about active-layer, calculated based on the thermal data? Or thaw depth measured by mechanical probing, that could be or not coincident with the active layer depending when do you measure it in the process of thawing of the ground? If I understooth correctly, the study sites have boreholes, right? Why not to calculate the real active layer thickness by the temperatures?
The depth of thawing is determined by mechanical sounding once a year during the maximum thawing of the soil in mid-September. In other words, this is the thickness of the active layer. To calculate the thickness of the active layer by temperature, it is necessary to install thermal sensors, which are short-lived and often fail. Therefore, preference is given to directly determining the thickness of the active layer.
Line 381: interannual variability or long-term variability?
Long-term variability.
Line 384: Again, where can I see the results of the statistics?
To avoid discrepancies in the text of the manuscript, the word "statistically" has been removed. “In the shallow-valley type of terrain at Sites 3a and 8, a significant increase in the depth of seasonal thawing was noted. Significant tendencies of its decrease are observed on sites of slope type of terrain (Sites 5, 6b, 9, 11). And if in the watershed area (Site 9) this decrease can be attributed to the intensive growth of shrubs, then on Sites 5 and 6b, most likely, fluctuations in the level of permafrost waters of the seasonally thawed layer play a significant role. At other sites (7b, 10), with significant interannual changes in the depth of seasonal thawing, insignificant tendencies of its growth are observed (Table 6).”
Lines 390-391: I do not know how to interpret the last column.
Trends (sm/10 year)
In summary, the mauscript and the science inside is very interesting and the data seems to be impresive, but i think that the manuscript need small improvements to help the reader to follow all the ideas presented inside, and the final conclusions.
It is a very interesing manuscript that i would like to see publish after its minor revision. Well done!

Round 2
Reviewer 1 Report
I appreciate the effort of the authors for revising the manuscript. I am happy to see the study area in the revised manuscript. However, the manuscript still lacking a few important things for getting published. The authors were suggested to cite the energy balance equations, which are not revised yet. Please provide the reference for the equations.
Similarly, it is not clearly mentioned the difference between past studies and this study in the same region. Please try to give specific significance and contribution of this study.
Discussion and usefulness of this study need to be emphasized. The result and discussion chapter is mainly focused on results (data, statistics). Please add some importance to those statistics.
Please also check the format of manuscripts, some tables are divided into two pages.
With these improvements, I think the manuscript will be suitable for publication in Journal Land.
Author Response
Response 1:
Point 1: The authors were suggested to cite the energy balance equations, which are not revised yet. Please provide the reference for the equations.
Компоненты тепло- и влагообмена земной поверхности с атмосферой выражаются уравнением теплового баланса или уравнением энергетического баланса на поверхности Земли [10].
Смотрите строки 122-123 в рукописи.
Пункт 2: Точно так же не ясно упоминается разница между прошлыми исследованиями и этим исследованием в том же регионе. Пожалуйста, попробуйте дать конкретное значение и вклад этого исследования.
Изменения теплового состояния верхних горизонтов вечной мерзлоты в Центральной Якутии и природных ландшафтов за последние 30-40 лет рассматривались на разных этапах исследований в [19-22, 33].
См строки 62-64 в рукописи.
Пункт 3: Обсуждение и полезность этого исследования должны быть подчеркнуты. Глава о результатах и обсуждении в основном сфокусирована на результатах (данные, статистика). Пожалуйста, добавьте некоторую важность к этой статистике.
Таким образом, изменчивость основных климатических параметров (температуры воздуха, осадков, снежного покрова) за последние десятилетия проявляется по-разному. Положительная динамика среднегодовой температуры воздуха является наиболее значимой. Осадки и высота снежного покрова подвергались кратковременным колебаниям без четкой тенденции.
См строки 201-204 в рукописи. Считаем достаточным резюме для подразделов 3.2 (строки 284-291), 3.3 (строки 361-367) и 3.4 (строки 399-408).
Пункт 4: Пожалуйста, проверьте формат рукописей, некоторые таблицы разделены на две страницы.
Да, исправлено.